# An Approach to Binary Classification of Alzheimer’s Disease Using LSTM

**DOI:** 10.3390/bioengineering10080950

**Published:** 2023-08-09

**Authors:** Waleed Salehi, Preety Baglat, Gaurav Gupta, Surbhi Bhatia Khan, Ahlam Almusharraf, Ali Alqahtani, Adarsh Kumar

**Affiliations:** 1Yogananda School of AI, Shoolini University, Bajhol 173229, India; waleedsalehi@shooliniuniversity.com (W.S.); gaurav@shooliniuniversity.com (G.G.); 2Interactive Technologies Institute (ITI/LARSyS and ARDITI), University of Madeira, 9000-082 Funchal, Portugal; pbaglat36@gmail.com; 3Department of Data Science, School of Science, Engineering and Environment, University of Salford, Manchester M5 4WT, UK; surbhibhatia1988@yahoo.com; 4Department of Business Administration, College of Business and Administration, Princess Nourah Bint Abdulrahman University, P.O. Box 84428, Riyadh 11671, Saudi Arabia; aialmusharraf@pnu.edu.sa; 5Department of Networks and Communications Engineering, College of Computer Science and Information Systems, Najran University, Najran 61441, Saudi Arabia; asalqahtany@nu.edu.sa; 6School of Computer Science, Mohammed VI Polytechnic University, Ben Guerir 43150, Morocco; 7School of Computer Science, University of Petroleum and Energy Studies, Dehradun 248007, India

**Keywords:** long-short-term-memory, magnetic resonance imaging, Alzheimer’s disease, deep learning

## Abstract

In this study, we use LSTM (Long-Short-Term-Memory) networks to evaluate Magnetic Resonance Imaging (MRI) data to overcome the shortcomings of conventional Alzheimer’s disease (AD) detection techniques. Our method offers greater reliability and accuracy in predicting the possibility of AD, in contrast to cognitive testing and brain structure analyses. We used an MRI dataset that we downloaded from the Kaggle source to train our LSTM network. Utilizing the temporal memory characteristics of LSTMs, the network was created to efficiently capture and evaluate the sequential patterns inherent in MRI scans. Our model scored a remarkable AUC of 0.97 and an accuracy of 98.62%. During the training process, we used Stratified Shuffle-Split Cross Validation to make sure that our findings were reliable and generalizable. Our study adds significantly to the body of knowledge by demonstrating the potential of LSTM networks in the specific field of AD prediction and extending the variety of methods investigated for image classification in AD research. We have also designed a user-friendly Web-based application to help with the accessibility of our developed model, bridging the gap between research and actual deployment.

## 1. Introduction

Human life expectancy has increased worldwide as a result of improved diagnosis and treatment. However, neither medication nor treatment can cure Alzheimer’s disease. More than 47 million people worldwide are affected by the condition [1]. Furthermore, the number of people affected by the condition is increasing yearly, and deaths due to the disease are increasing, while deaths from other diseases are declining. In the United States, for example, the number of people affected by the disease is anticipated to increase to 13.8 million by 2050, up from 5.4 million in 2016. The rate of diagnosis is expected to more than double by 2050, from 66 s in 2016 to 33 s. Alzheimer’s disease was responsible for 84,767 fatalities in 2013. Between 2000 and 2013, mortality from chronic diseases such as stroke and heart disease declined, whereas deaths from the disease surged by 71% [2].

In the preliminary stages of AD, common obstacles include [3]:It might be challenging to recall proper words or names.Daily work in social settings or an office might be difficult.Getting in trouble remembering people’s names while meeting them.Having difficulty locating or misplacing a valuable asset.Forgetting something you just heard or read.It might become much more difficult to plan and manage various daily tasks.

Recent developments in deep learning algorithms and other Machine Learning (ML) techniques have made it possible to extract relevant data for data classification across a variety of disciplines. Due to their outstanding performance in image processing and computer vision applications, deep learning (DL) techniques offer a wide range of potential applications [4]. The process of identifying AD in its early stages can be both costly and time-consuming, according to past research on disease detection and forecasting. This entails gathering a large quantity of data, using sophisticated diagnostic techniques, and consulting an experienced clinician [5]. Because they are susceptible to human mistakes, computerized mechanisms are more accurate than human evaluations and can be used in medical systems [3]. MRI images [6], because of their capability of showing the anatomical makeup of the brain, have been shown in the literature to be a useful method for diagnosing AD [7]. The ability of deep learning approaches to manage large, complex data sets, including high-dimensional and multi-modal data, is a key advantage. For instance, DL-based techniques can be used to examine the intricate connections between brain function and structure and the onset of AD when using MRI data to make such predictions. RNNs, CNNs, and LSTM algorithms are a few deep learning techniques that have been used to predict AD [8]. These methods have been used to the analyze clinical data, genetic information, MRI data, and results from cognitive tests. The accuracy of disease diagnosis could be greatly improved by using deep learning algorithms for disease prediction and better patient outcomes and quicker treatment would result. These techniques can also provide new perspectives on the underlying causes of illnesses, which can help with the creation of more potent remedies and preventative measures [9]. 

Although there is no cure for AD and a definitive diagnosis is difficult, early detection is essential for preventing symptoms from getting worse [10]. Through early identification, AD patients’ quality of life can also be enhanced [11]. In order to automatically diagnose AD, research has been done on DL computer-aided [12] techniques [13]. Both CNNs and RNNs are strong and reliable varieties of artificial neural networks. Another prominent form of RNN-based algorithm is known as LSTM, which is one of the most promising algorithms currently in use since it has internal memory [14]. Like many other DL techniques, we have seen the true potential of Recurrent Neural Networks. RNN-based methods are now more popular than ever thanks to improvements in technology, greater data accessibility, and algorithms like LSTM. Due to its capacity for dealing with difficulties like exploding and vanishing gradients, LSTM is favored over conventional RNNs. Its adaptability and memory capacity make it perfect for jobs like time series analysis, speech recognition, and NLP, which promotes better understanding in the data-driven environment [15].

In this study, we strive for a time-series-based technique known as LSTM. The LSTM model receives the MRI images as input, and after performing feature extraction based on each image’s pixel values, the model outputs a binary classifier that divides the input into one of two categories, such as Non-Demented or Very-Mild-Demented. The key main aim of the paper is as follows:Objective 1:Develop a robust and accurate diagnostic LSTM approach and utilize its capacity for time-series MRI data analysis.Objective 2:Investigate how well the suggested LSTM-based model performs in categorizing MRI data and making accurate predictions for the early identification of AD.Objective 3:To offer a thorough examination of the model’s diagnostic skills and evaluate the model’s efficacy using a wide range of measures, such as accuracy, confusion matrix, and AUC ROC.Objective 4:Create a user-friendly Web-based application to aid in the practical deployment of the generated model.

The remainder of the paper is divided into numerous sections, beginning with the introduction, which gives a background on the subject being studied and a summary of LSTM Approach. The next section, the methodology, describes the research strategy, including the methods used for data collection and analysis. The literature review comes next. The findings from the study and evaluation are presented in Section 5.2, exploratory data analysis is covered in Section 5, and future extensions and research implications are discussed in the paper’s conclusion.

## 2. Overview of Deep Learning and LSTM Model

Complex Artificial Neural Networks (ANNs) are used in DL, a subset of ML, to find patterns in data that may be difficult to understand. Deep learning algorithms are based on how the human brain is built and how it works. They can be used to do things like recognize images [16] and voices, process natural languages, etc. [17]. Unlike machine learning, models that are based on deep learning are made up of numerous layers, each of which learns progressively abstract representations of the data that are fed into the model. Early layers are responsible for the acquisition of fundamental characteristics, such as edges and color blobs; later layers, however, are in charge of integrating these characteristics to recognize objects, predict outcomes, and make choices. DL algorithms can be broken down into a few distinct types, the most common of which are feedforward NNs, Recurrent Neural Networks (RNNs), and Convolutional Neural Networks (CNNs). The particular issue at hand and the nature of the data to be entered both play a role in the model selection process. CNNs are an artificial neural network type developed for image-identification tasks. They filter picture inputs using convolutional layers and extract important features. Pooling layers are then employed to minimize the spatial dimensions of the filtered output and keep only the most essential data. The retrieved features are then transmitted through layers with complete connectivity to get the final classification results. In computer vision applications, CNNs have achieved great success and are frequently used for image classification, object recognition, and picture segmentation. On the other hand, LSTM, which was introduced by Hochester and Schmidhuber in 1997, is a specialized type of RNN that was developed to address the issues of exploding and vanishing gradients in traditional RNNs [18]. These issues arise when training RNNs on long sequences, making it difficult for the network to retain and propagate information over time. LSTMs mitigate these problems by introducing a more complex and sophisticated architecture. They incorporate memory cells, which are capable of storing information for extended periods, allowing the network to retain important information over long sequences [19]. Additionally, input, forget, and output gates are included in LSTMs to control the information flow into, out of, and within the memory cells [20]. These gates are responsible for determining whether to accept further input (input gate), eliminate the data since they are irrelevant (forget gate), or permit the information to influence the output at the current time step (output gate). Additionally, the cell unit memory keeps values over a series. Pointwise multiplication operations and nonlinear functions are what make up these gates.

LSTMs have a topology similar to a chain, but their repeating module is constructed differently. Instead of just one, there are four, and their interactions are highly unusual. Each line in Figure 1, which shows a high-level perspective of the LSTM architecture, transports a full vector from one node’s output to another node’s input. The pink circles in the notations represent pointwise operations such as vector multiplication and addition, and the yellow notations represent trained neural network layers. Concatenation is indicated by line merging, but a forked line indicates that its content has been replicated and is being sent to other destinations [21]. Following are descriptions of each gate in LSTM architecture:Forget Gate—The input and previous output are combined at the forget gate to produce and generate an output that falls within the range of 0 to 1, and a sigmoid activation function is employed, which indicates how much of the previous timestamp has to be kept or forgotten. The previous state is multiplied by this output after that [22].Input Gate—Although the input gate’s output ranges from 0 to 1, it operates with the same signals as the forget gate. The goal here is to choose which new information will be added to the LSTM’s state [18]. The additional values that were to be added to the initial state are then created by multiplying them by the output of the tan h block. The current state is created by adding this gated vector to the previous state.Output Gate—The output gate will gate the input and previous state as before to produce an output that ranges from 0 to 1, which is then combined with the output of the tan h block to obtain the current state. The result is then distributed [22]. The LSTM block receives input from both the output and the state.

### Importance of Imaging Modality for AD Detection

MRI is a non-invasive medical imaging method that creates images of inside body components. This method creates detailed images of the body’s organs, tissues, and bones using a computer, radio waves, and a strong magnetic field [23]. According to the recent literature [24], MRI has been the most popular diagnostic technique for detecting AD which is shown in Figure 2 due the distribution of usage of various imaging modalities. AI is increasingly being used in MRI technology to improve various aspects of the imaging process, including.

Image analysis: AI algorithms can be used to analyze MRI images and detect anomalies or patterns in including. allowing for earlier and more accurate diagnoses.Image reconstruction: AI can be used to enhance MR picture quality by reducing noise and boosting resolution.Image segmentation: Radiologists can more easily identify particular areas of interest by using AI to segment MRI pictures into various tissues or structures.Dose reduction: AI can be used to optimize MRI protocols to reduce the amount of radiation exposure for patients, making the imaging process safer [25].Personalization: AI can be used to personalize MRI scans for individual patients based on factors such as body shape, size, and medical history.

## 3. Related Works

The field of DL has recently appeared as a potentially useful method for the diagnosis and prognosis of diseases such as AD, a neurodegenerative disorder that impacts millions of people globally and worsens over time. Deep learning algorithms can detect subtle patterns and alterations in the brain that might signify the initial phases of AD. This is accomplished by evaluating enormous volumes of data, which may include scans of the brain as well as genetic information. Several studies utilizing CNNs, RNNs, and LSTM algorithms have been published on the subject of deep-learning-based strategies for detecting AD. This research has made significant contributions to the enhancement of novel and more effective AD diagnostic and treatment procedures.

Some of the key findings and contributions of previous studies in this field include:Temporal information—LSTMs have been used in multiple research projects to incorporate temporal information into the prediction model, which can help us understand Alzheimer’s disease and its progression better. LSTMs are particularly well-suited for studying time-series data [26].Improved accuracy—CNNs and transfer learning have been shown in multiple studies to attain a high level of accuracy when it comes to the prediction of Alzheimer’s disease, in contrast to more traditional techniques of machine learning [27]. This could lead to earlier, more accurate diagnoses and better patient outcomes [27].Transfer learning—Transfer learning has been utilized in some research to make use of CNN and LSTM models that have already been trained on massive amounts of data and can then be tailored to particular tasks and data sets. This has the benefit of requiring less data and processing during training, which can increase prediction accuracy [28].Integration of multi-modal data—Several studies have combined the results of MRI scans with those of other clinical assessments, such as cognitive evaluations or genetic information. This has resulted in a more in-depth and precise understanding of Alzheimer’s disease and how it progresses [29].

In recent years, there has been a lot of interest in the application of RNN-based approaches for the prediction of Alzheimer’s disease. In this literature review, we explore six recent papers that have used various RNN architectures for AD prediction. A Multi-Layer Perceptron (MLP) and a Bidirectional Gated Recurrent Unit (BGRU) were used to classify AD, done by the author in 2018 [10]. The authors of [26] employed LSTM to predict the future state of AD in 2019. A CNN was utilized to learn spatial characteristics in the third paper [30], and an RNN was used to extract longitudinal features for classification. Another piece of work [31] from 2020 used an ensemble method to combine CNN, RNN, and LSTM models to reach high accuracy. The fifth study [12], released in 2021, proposed a complete 3D framework based on ConvLSTM for early AD diagnosis. The last study employed LSTM to predict biomarkers and a neural network for classification in [32], which was published in 2022. Overall, these findings indicate that RNN-based methods for predicting AD have promising outcomes. These models’ levels of accuracy range from 86% to 92.22%. The generality of these models and their application to bigger datasets could be improved with future research, though. In conclusion, RNN-based algorithms have considerable promise for AD diagnosis and prediction, and additional study in this area has the potential to significantly advance science. 

Deep learning algorithms have attracted a lot of interest recently and have demonstrated incredible promise for the detection and analysis of brain disorders. These algorithms have been extensively studied and applied to numerous other brain illnesses besides Alzheimer’s disease, significantly advancing medical imaging and diagnosis. The diagnosis of Autism Spectrum Disorder (ASD) is a significant use of deep learning algorithms [33]. ASD is a complicated neurodevelopmental disease marked by issues with social interaction and communication. Deep learning algorithms have been used to analyze neuroimaging data from structural and functional MRI to find patterns and biomarkers connected to Autism Spectrum Disorder (ASD). These models have demonstrated promise for increasing the precision of ASD diagnosis and comprehending the underlying variations in brain connections in people with ASD [34].

In the diagnosis of schizophrenia (SZ), DL algorithms have also demonstrated substantial promise. SZ is a persistent mental illness marked by abnormal perceptions, actions, and thoughts. Deep learning algorithms have been used to identify useful characteristics and patterns suggestive of SZ from multimodal neuroimaging data, including fMRI, diffusion tensor imaging (DTI), and electroencephalography (EEG) [35]. These models have proven to be effective at differentiating between healthy people and people with SZ, facilitating early detection and individualized treatment plans.

For instance, utilizing resting-state functional MRI (rs-fMRI) data, Shoeibi et al. [35] present a new DL strategy for the intelligent detection of schizophrenia (SZ) and attention deficit hyperactivity disorder (ADHD). In their method, the data are preprocessed, features are extracted using a convolutional autoencoder model, and interval type-2 fuzzy regression (IT2FR) using optimization approaches is used. The IT2FR approach outperformed other classifier methods with an accuracy rate of 72.71%. The study emphasizes the potential of deep learning and fuzzy regression for rs-fMRI data-based SZ and ADHD detection. By combining neuroimaging modalities, another paper [36] investigates the potential of deep learning models for diagnosing brain diseases. It looks at different models, including CNNs, RNNs, GANs, and AEs, and analyzes the benefits of deep learning over more traditional approaches. Deep learning algorithms have also been used to analyze different types of brain conditions, including Parkinson’s disease [37], epilepsy [38], and brain cancers [39]. These algorithms have demonstrated promise in identifying patterns particular to certain diseases, assisting in accurate diagnosis, and offering insights into the progression of the disease and the effectiveness of treatment.

For instance, two CNN frameworks are shown by Hakan Gunduz et al. [37] for identifying Parkinson’s disease based on vocal (voice) data. How the frameworks combine feature sets varies. A nine-layered CNN is used in the first framework to merge various feature sets. The second architecture, in contrast, sends feature sets to parallel input layers that are coupled with convolution layers directly. As a result, deep features can be simultaneously extracted from each parallel branch before being combined into the merged layer. The suggested models were tested using Leave-One-Person-Out Cross Validation (LOPO CV) using a dataset from the UCI Machine Learning repository.

Two deep learning algorithms for the detection and classification of brain tumors are introduced in another study [40]. The You Only Look Once (YOLO) object-identification framework is used in the first technique, and the FastAi deep learning library is used in the second. The study focuses on a subset of 1992 MR brain images from the 2018-BRATS dataset. The accuracy of the YOLOv5 model is 85.95%, whereas the accuracy of the FastAi classification model is 95.78%.

In conclusion, deep learning algorithms have become effective instruments for identifying and analyzing brain disorders. Using neuroimaging data from structural and functional MRI, they have been effectively applied to several brain illnesses, including AD, ASD, and SZ. These algorithms have shown increased diagnostic accuracy and given insights into the underlying abnormalities in brain connections that underlie these illnesses. Deep learning methods have also demonstrated promise in the detection of various neurological conditions such as Parkinson’s disease, epilepsy, and brain tumors, enabling accurate diagnosis and comprehension of disease progression. Diagnostic abilities are further improved by combining deep learning with cutting-edge approaches like fuzzy regression and multimodal fusion. Overall, the use of deep learning algorithms for the diagnosis of brain disorders holds tremendous promise for enhancing individualized care plans and deepening our understanding of these challenging ailments. A summary of the studies discussed above is given in Table 1.

## 4. Materials and Methods

### 4.1. Data Description

Our dataset was sourced from an online Kaggle challenge, specifically focusing on MRI brain images. The dataset provided for this challenge included a total number of 6400 images, including training and testing and it is available into four distinct classes, Non-Demented, Mild-Demented, Moderate-Demented, and Very-Mild-Demented; and contained 200 subjects, with 32 slices of the image for each subject. The data source can be found in Kaggle (Alzheimers-dataset-4-class-of-images) [41]. Out of these four classes, we only considered Non-Demented and Very-Mild-Demented since the implemented model is a binary classifier.

For the proposed approach, we used two groups of data, such as Non-Demented and Very-Mild-Demented. Table 2 provides an overview of the distribution of Non-Demented and Very-Mild-Demented MRI scans specifically for training and testing. This table serves to present the two distinct classes of MRI scans, namely Non-Demented and Very-Mild-Demented, and highlights the number of data samples allocated for training and testing within each class. By examining the table, one can easily discern the partitioning of data for both the training and testing phases, facilitating a clear understanding of the dataset distribution across different categories of MRI scans.

### 4.2. Proposed Method

Alzheimer’s disease has been predicted using DL-based approaches in many cases. These techniques use ANNs, which are designed to learn patterns and relationships in data. By training these networks on large amounts of data, they can learn to recognize complex relationships and make accurate predictions [42]. The two primary stages of training a DL model are forward propagation and backward propagation, which are used to calculate the loss function between the predicted output and the ground truth labels [20]. In this kind of DL model, the main aim is to reduce the loss as much as possible in such a way that the expected output gets closer to the actual output. Convolutional neural networks, one sort of DL-based approach, have become quite popular in image classification [43] over the past few years and have been employed extensively for a range of issues such as image segmentation, classification [44,45], and other problems [46]. These types of models only work with numbers, but they are unable to comprehend images the same way that humans do. To make the computer understand the images, we must somehow translate them into numbers [47], then the model can extract meaningful features and give output based on the observed features. We used MRI images and preprocessed them into desirable input for the model. The idea here is we are passing the image’s rows as sequences in this network. In our study, we utilized MRI images as the input data for our model. To prepare the images for the model, we performed preprocessing steps to ensure they were in the desired format. One key aspect of our approach was to consider the rows of the image as sequences within the neural network. In other words, we treated each row of pixels in the image as a separate sequence, allowing the model to process the image sequentially. This approach resulted in an input shape of 100 × 100, indicating that we had 100 sequences, each consisting of 100 elements. In other words, 100 rows (sequences) contain 100 columns (pixels). To incorporate the sequential nature of the data into our model, we included an LSTM layer. This layer had a parameter called “return sequences”, which served as a flag indicating whether the model should continue to another LSTM recurrent layer or not. By enabling this parameter, we allowed the model to capture and learn from the sequential patterns present in the image data. Under the proposed approach, the following steps were taken and concisely discussed:Image Preprocessing—The CV2 library was utilized for image preprocessing. The images were resized, and labels were appended to them.Normalization—The image data were normalized by dividing them by 255.0 to scale the pixel values between 0 and 1.Categorical Target—The np_utils module from Keras was employed to convert the target labels into categorical format.Model Building—The TensorFlow and Keras frameworks were utilized. The necessary modules were imported to construct the model. The model consisted of multiple layers, with the CuDNNLSTM layer serving as the backbone. The layers were added sequentially, employing the he_uniform kernel initializer and setting the return_sequences attribute to true. Dropout layers with a rate of 0.2 were added between each layer to avoid overfitting. The ReLU activation function was applied to the layers. For the final layer, a sigmoid activation function was used since it was a binary classification task.Stratified Shuffle-Split Cross-Validation—The Stratified Shuffle-Split Cross-Validation technique was employed with 5 splits and a test size of 0.1. This technique ensured the preservation of the class distribution in the training and testing datasets.Model Compilation—The model was compiled with parameters such as a loss function set to binary_crossentropy, metrics for evaluation, and an optimizer. The optimizer chosen was SGD (Stochastic Gradient Descent), and a grid search approach was used to find the best optimal values by trying different learning rates and momentums.Early Stopping—Early stopping was put into place to terminate training as soon as the model’s performance on the validation set began to deteriorate to avoid overfitting.Best Fold Selection—The best fold (score) was determined using the arg max function, and the corresponding best accuracy was obtained.

The model’s performance was then plotted and visualized using a variety of metrics, including Accuracy, Confusion Matrix, Area Under the Receiver Operating Characteristic (ROC) curve (AUC ROC), and accuracy scores for cross validation.

Lastly, to enhance user accessibility, a Flask application was created to enable individuals to input their own MRI images and receive prediction results from the deployed model. To do this, the trained model was saved and imported into the Flask application. When a user submits an image, the application uses the imported model to generate a prediction for the input image.

LSTM models were specifically developed to tackle the long-term dependency issue in RNNs caused by the vanishing gradient problem. In contrast to traditional RNNs, LSTM models mitigate this problem by incorporating specialized memory units that can retain and propagate information over longer sequences. This advancement in LSTM architecture has significantly improved the ability of recurrent neural networks to capture and model complex temporal relationships. LSTM is different from traditional forward neural networks, and it is the feedback connections that make it different. This property allows LSTMs to evaluate whole data sequences, such as time series, without having to take into account each data point separately. Instead, they keep track of relevant information from previous data points to assist in processing incoming data [48].

In the last layer for binary classification, a sigmoid function was used together with an ReLU activation function during training. The proposed workflow is shown in Figure 3. The training and validation carried out in this experiment served to both train the deep learning model and evaluate its performance on new data. For the model to learn from the data, the weights and biases were updated using the training dataset. A loss function was used during training to determine the difference between the model’s predictions and the actual target values. Minimizing the loss function during training means that the model was successfully fitting the data. Utilizing the validation dataset, the model’s performance was assessed following training [49]. Based on the validation data, the model was used to create predictions, and metric [50,51,52] accuracy was used to evaluate the model’s performance. The two phases such as model building and deployment flow are given in Figure 4.

Table 3 showcases the major hyperparameters used in the proposed model along with their corresponding values. These hyperparameters governed the behavior and performance of the model during training and inference. The table provides a concise summary of the specific values chosen for each hyperparameter and their purpose in the model.

The hyperparameters included in the table encompass important aspects such as the dropout rate, batch size, activation functions, accuracy as the evaluation metric, loss function, optimizer, learning rate, momentum, number of epochs, and early stopping. Each of these hyperparameters contributed to different aspects of the model’s training and optimization process.

## 5. Experiment and Evaluation

### 5.1. Setup for Experiment

The DL model in this study was run on an Intel Computer OptiPlex 7090 GeForce RTX 3070 system, and the tests were conducted using the TensorFlow framework. The LSTM model for the prediction of AD using MR images was trained and effectively evaluated thanks to the utilization of a dedicated GPU and the TensorFlow framework. The TensorFlow framework provided a flexible and powerful platform for building and training deep learning models, and the dedicated GPU provided the necessary computational resources to handle the large amounts of data and complex computations involved in the experiment. The entire code was implemented in Python programming language. For training, the CUDA driver was utilized and endorsed by NVIDIA for graphics processing. Version 9.0.176 of CUDA was used to optimize the use of GPU resources.

### 5.2. Results and Discussion

To assess the effectiveness of the suggested DL approach for the prediction of AD, experiments were carried out. The performance of the LSTM model was assessed after it was trained using MR images from the two categories of Non-Demented and Very-Mild-Demented. The experiment presented in this research offers the following outcomes: accuracy of 98.62%, loss of 0.003, validation loss of 0.041, and validation accuracy of 0.976, demonstrating the promise of DL-based techniques like LSTM networks in the diagnosis of AD. The proposed approach provides a promising solution for the early and accurate prediction of AD using MRI brain scans, and the results show that the LSTM model can be used to improve the current methods of AD diagnosis and aid in the development of effective treatment strategies. Researchers can better understand the model’s operations and identify areas for improvement by examining the behavior of the model and displaying the change in accuracy as a function of the number of epochs. The accuracy of the model is plotted versus the number of training epochs in Figure 5, with each epoch denoting a full loop through the training dataset. The y-axis shows the accuracy of the model, which can range from 0 to 1, and the x-axis represents the number of epochs, which grows from left to right. The graphic shows that, typically, as the number of epochs rises, so does the model’s accuracy. This is a result of the model gradually improving its predictions over time by learning from the input data.

The loss of the model is displayed versus the number of training epochs in Figure 6. The graph demonstrates that as the number of epochs rises, the model’s loss decreases. As a result, the model’s predictions and the actual target values are becoming closer together as a result of the model’s gradual learning from the data.

In our investigation, our model produced an amazing AUC of 0.97. This high AUC shows how well our model performs in differentiating between the positive and negative classifications. By examining the ROC curve, we discovered that, with a probability of 0.97, our model consistently scored a randomly chosen positive case higher than a randomly chosen negative instance. Strong support for the efficacy and dependability of our model in correctly classifying the target variable is provided by the achieved AUC of 0.97. This exceptional result highlights the robustness and discriminative strength of our model in identifying the underlying patterns and producing precise predictions, as illustrated in Figure 7. The TPR (True Positive Rate) examines how well a classifier can detect positive examples, whereas the FPR (False Positive Rate) shows how often false alarms are generated. TPR measures sensitivity, and FPR gauges the classifier’s specificity. With high TPR and low FPR serving as the optimal point on the ROC curve, we can evaluate the trade-off between sensitivity and specificity. The performance of the classifier is summarized by the AUC ROC.

The confusion matrix for the classification of AD is shown in Figure 8. The performance of the classification model in determining an individual’s disease-state based on a set of features is shown in the image. The matrix, which summarizes the number of true positives, false positives, true negatives, and false negatives, is a tabular form with two rows and two columns. The illustration emphasizes the significance of correctly diagnosing Alzheimer’s patients, because false negatives may cause treatment and diagnosis to be delayed. Briefly, “0” denotes those without Alzheimer’s disease and “1” denotes people who have the disease.

Figure 9 displays the outcomes of our thorough study using Shuffle-Split Cross-Validation and highlights the outstanding performance of our model. The y-axis shows the various folds used, with a total of five folds, while the x-axis shows the accuracy levels attained. Surprisingly, across all folds, our model consistently exhibits outstanding accuracy. The third fold, in particular, stands out as the pinnacle of performance with a remarkable accuracy of 98.62%. This outstanding outcome demonstrates how well the model can identify and categorize intricate patterns in the dataset. The dotted red line’s representation of the average mean accuracy further supports the model’s dependability and resilience. The model’s ability to generalize well and retain good performance on various subsets of the data is shown by the consistently high accuracy scores throughout the folds.

Table 4 displays the dataset utilized for evaluation along with the performance results of several models.

A promising tool for medical diagnosis is the Web application created for AD prediction. It has the capacity to input MRI scans of patients and provide a user-friendly UI for the user. The Web program uses advanced algorithms to provide accurate predictions of Alzheimer’s disease, enabling early detection and treatment. The functionality and user interface of the application is illustrated by Figure 10, Figure 11 and Figure 12, given below.

The diagnosis and prognosis of Alzheimer’s disease have shown tremendous potential for deep learning approaches. Deep learning can identify small anomalies linked to the disease and capture complex patterns in neuroimaging data by employing models like CNNs and RNNs. These algorithms provide unbiased evaluations, early identification, and individualized risk evaluations. Deep learning has the potential to transform clinical practice by utilizing cutting-edge computational techniques, enhancing patient outcomes, and expanding our understanding of Alzheimer’s disease. The proposed model does, however, have a drawback due to the availability of a tiny dataset. The dataset’s small size may make it difficult to fully capture the complexity and variety of the target population. The generalizability and robustness of the model’s performance may be impacted by this.

## 6. Exploratory Data Analysis

An important stage in any data-driven endeavor is EDA (Exploratory Data Analysis). With the use of this technique, you may learn more about the data and spot patterns, trends, and anomalies that might not be obvious from summary statistics alone. In this section, we offer the dataset’s EDA results, which include feature visualizations and descriptive statistics. The descriptive statistics highlight the central tendencies and dispersion of the dataset, while the visualizations aid in the discovery of any links, trends, and patterns among the variables. Figure 13 shows the relative sizes of the two classes, with category one covering roughly 58.8% of the dataset and category two roughly 41.2%. These data emphasize the differential sample distribution between the two classes. It is significant to note that there might have been a little mismatch in the proportions of the classes due to the availability of the dataset. The deviation, nevertheless, is not significant enough to cause grave concerns. The dataset is still useful for training and testing the model, despite the modest disparity, because it includes a representative sample of both Non-Demented and extremely Mild-Demented cases.

Figure 14 shows that the mean pixel values of the Non-Demented and Very-Mild-Demented classifications varied noticeably. In particular, the Non-Demented class typically has a higher mean pixel value than the Very-Mild-Demented class. This implies that the two classes’ image colors differ from one another. Further information about the distribution of mean pixel values is also shown by the shape of the violins in the plot. The violin plot appears larger for the Non-Demented class, showing a greater range of mean pixel values. Additionally, the plot demonstrates that for the “non-demented” class, more values are distributed toward the higher end of the range.

The distribution of an image’s mean pixel values in relation to x-axis density is shown in Figure 15. The average pixel value on the y-axis, which displays the average color intensity of the image, is plotted against the pixel density on the x-axis. The graph makes it easy to see the color properties of the image as well as how the pixel values are distributed throughout its parts. 

The distribution of an image’s maximum pixel value according to its class—Very Mild-Demented or Non-Demented—is shown in Figure 16. The probability density of the maximum pixel value for each class is shown by the KDE plots in the picture. The likelihood or probability of seeing a specific maximum pixel value within each class is represented by smooth curves that are used to build the KDE displays. The probability that a particular value will occur within the appropriate class increases with the height of the curve’s peak or density at that value.

We can better comprehend the distribution patterns and variations in the maximum pixel values between the two classes by looking at the KDE graphs. If the curves’ peaks are at different locations or have distinct shapes, it means that the distributions of the classes’ maximum pixel values are not the same.

According to the statistical measurements employed to characterize the image samples, Figure 17 “showcases the average and variability of image samples, represented by their mean and standard deviation values”. The mean value indicates the typical or average image qualities within the collection by displaying the average or central tendency of the samples. Indicating the degree of variances or diversity present in the dataset, the standard deviation value shows the variability or spread of the samples around the mean. The picture provides a summary of the main image qualities and the range of variances seen in the dataset by displaying the mean and standard deviation values.

In the case of Figure 14, Figure 16 and Figure 17, the significant and distinctive relationship between pixel values and image samples is that pixel values relate to the numerical values that indicate the intensity or color of particular points in an image. We examine the distribution of mean pixel values across all classes in Figure 14. On the other hand, image samples are full images made up of several pixels. We look at the highest pixel value distribution per class in Figure 14. The mean and standard deviation values of the image samples that make up Figure 16 are used to illustrate the average and variability of those samples. We learn more about the traits and statistics of image samples as a whole by comprehending the pixel-level data contained in images.

## 7. Conclusions

We were able to successfully use a DL-based LSTM model to predict Alzheimer’s disease using MRI brain images, according to the findings of our study. With a remarkable AUC of 0.97, the model distinguished between the positive and negative classes with an accuracy of 98.62%, demonstrating both its extraordinary performance and its promise as a diagnostic tool for the disease’s early identification. The ability to accurately predict AD at an early stage can greatly benefit patients and healthcare professionals in planning appropriate interventions and therapies. Deep learning methods have demonstrated encouraging outcomes in several medical domains. This study adds to the growing corpus of research that shows these methods are effective when used in healthcare. A Web-based application was also designed to make the DL model easily accessible. It allows users to upload MRI scans, and the deep learning model provides a prediction of the likelihood of AD. Overall, our findings demonstrate the potential of the DL-based model to improve the accuracy and efficiency of disease diagnosis, and we believe that further research in this area can lead to significant advancements when it comes to medical imaging and diagnosis. As part of our future work, we aim to expand the dataset by increasing the number of samples. To achieve this, we plan to explore openly accessible repositories and apply data augmentation techniques. We want to increase the generalization of our model through this expansion of the dataset.

## Figures and Tables

**Figure 1 bioengineering-10-00950-f001:**
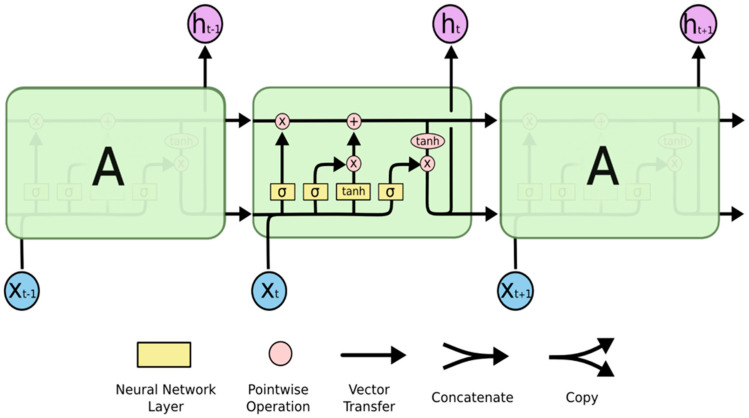
An LSTM has a repeating module with four interconnected layers [21].

**Figure 2 bioengineering-10-00950-f002:**
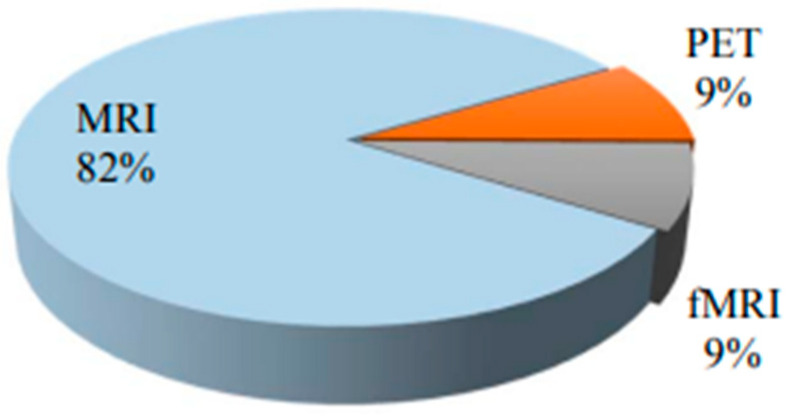
Usage of MRI as compared with other modalities.

**Figure 3 bioengineering-10-00950-f003:**
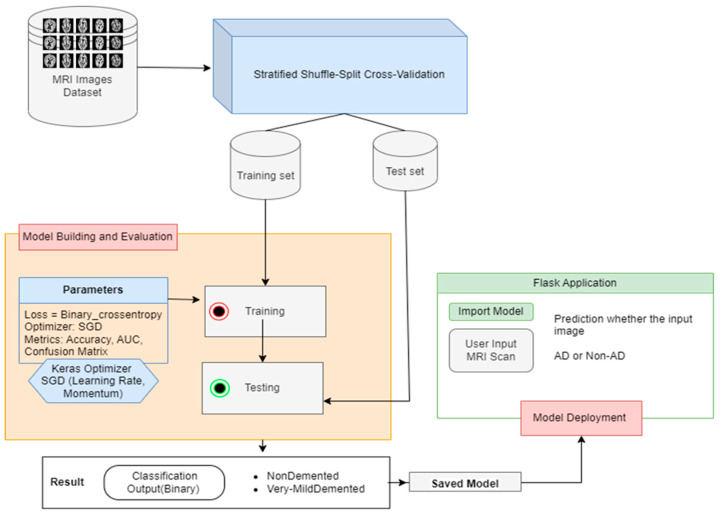
Proposed workflow diagram.

**Figure 4 bioengineering-10-00950-f004:**
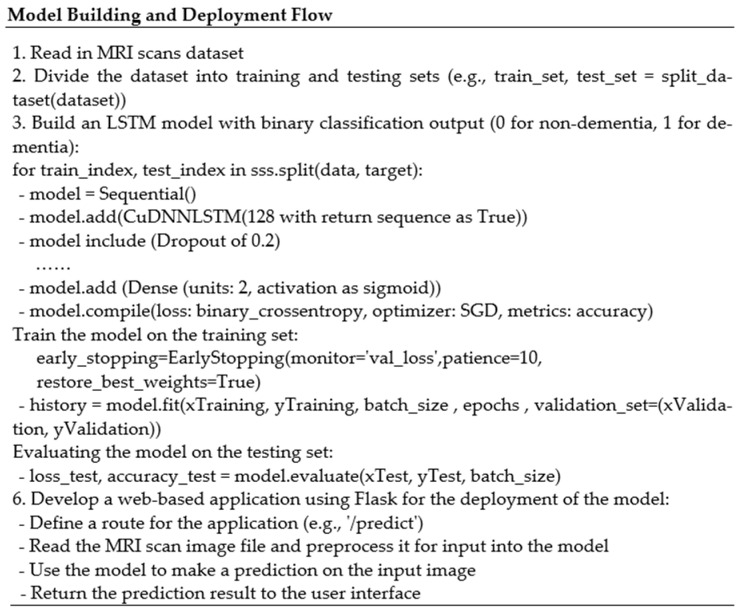
Model building and deployment.

**Figure 5 bioengineering-10-00950-f005:**
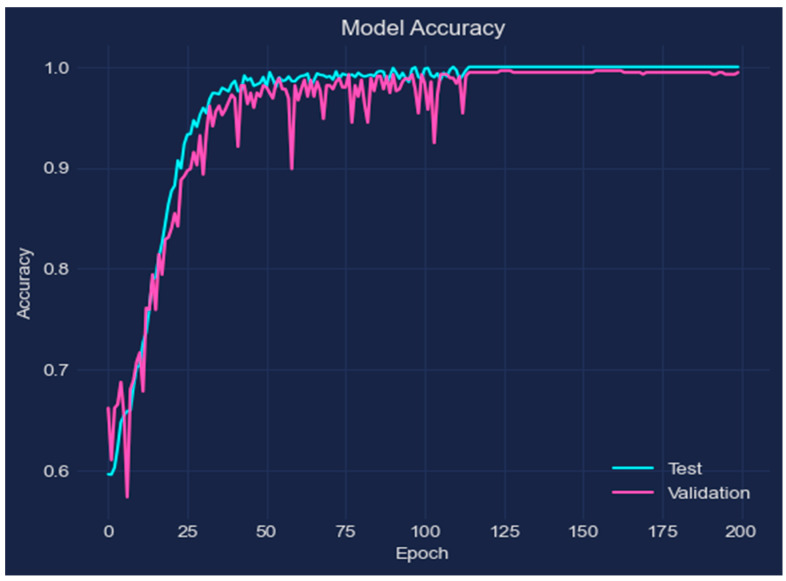
The performance of training accuracy.

**Figure 6 bioengineering-10-00950-f006:**
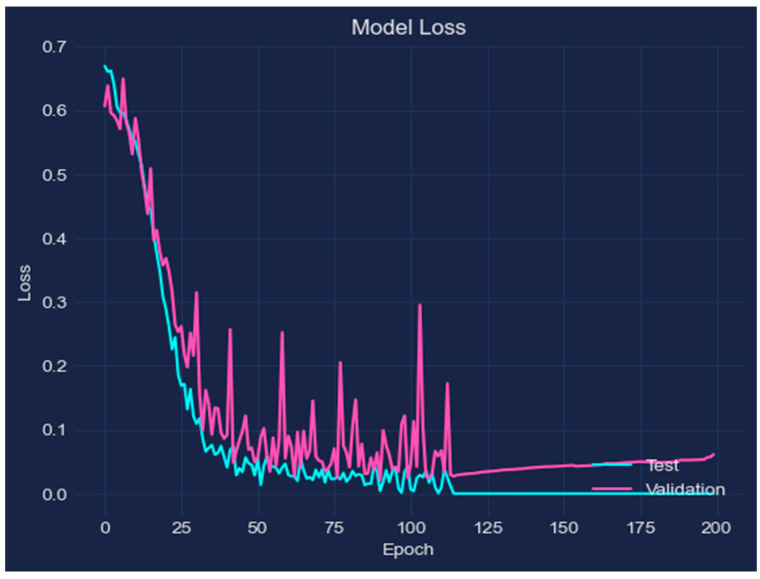
Loss of the model.

**Figure 7 bioengineering-10-00950-f007:**
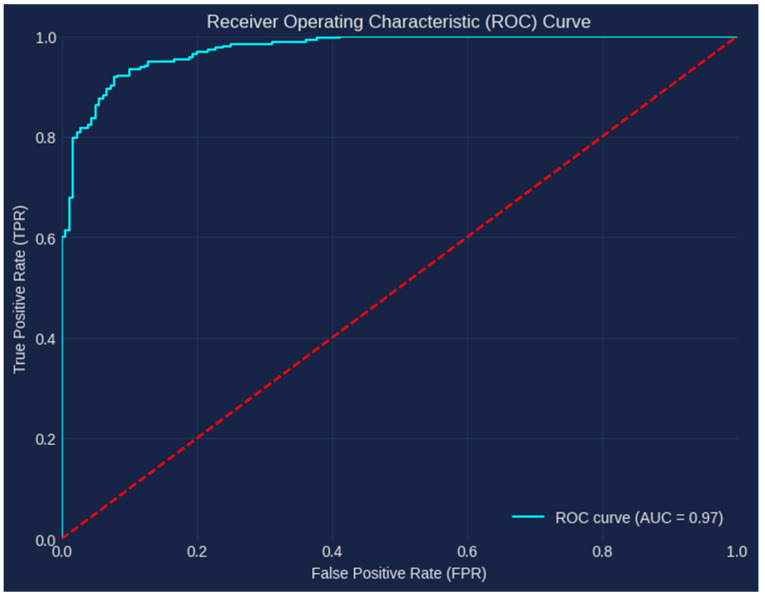
AUC ROC Curve.

**Figure 8 bioengineering-10-00950-f008:**
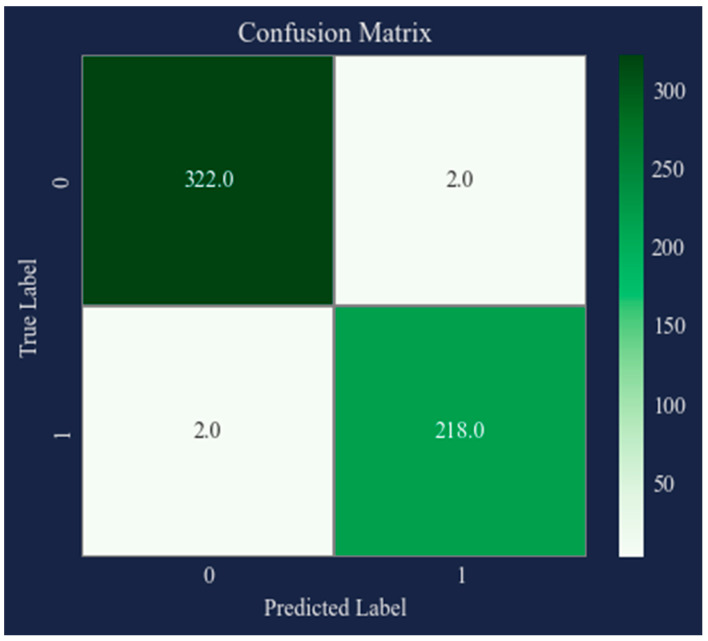
Confusion matrix.

**Figure 9 bioengineering-10-00950-f009:**
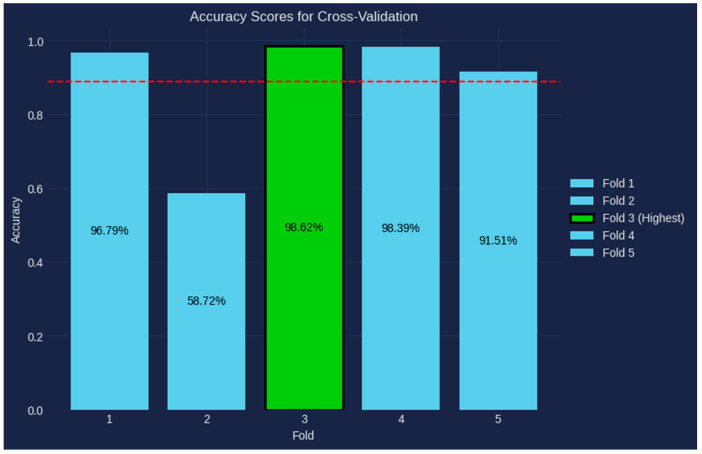
Accuracy scores for cross-validation.

**Figure 10 bioengineering-10-00950-f010:**
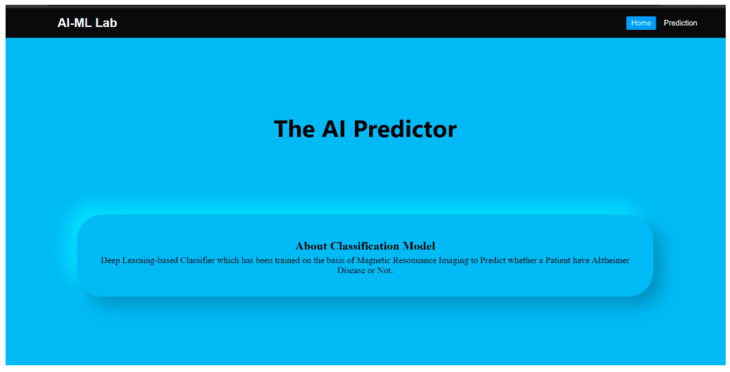
Home page.

**Figure 11 bioengineering-10-00950-f011:**
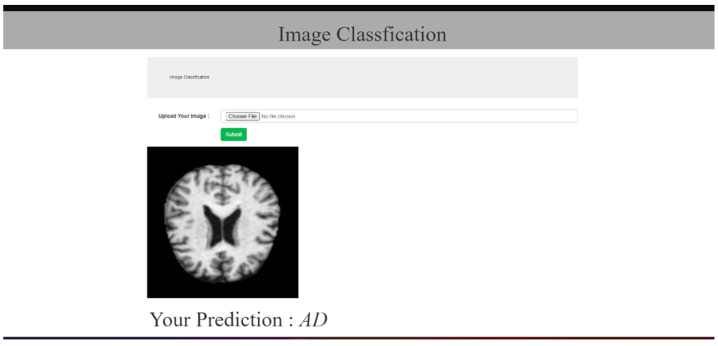
Prediction page (AD).

**Figure 12 bioengineering-10-00950-f012:**
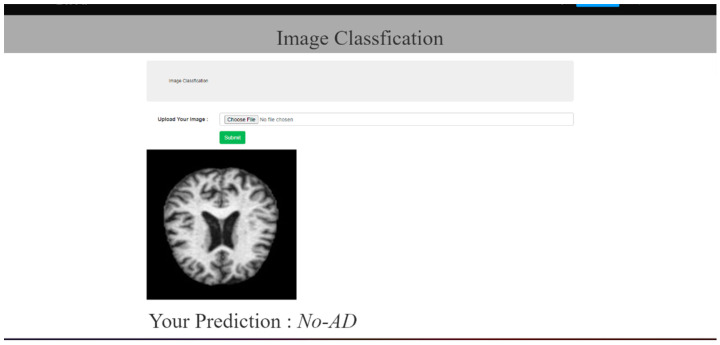
Prediction page (No-AD).

**Figure 13 bioengineering-10-00950-f013:**
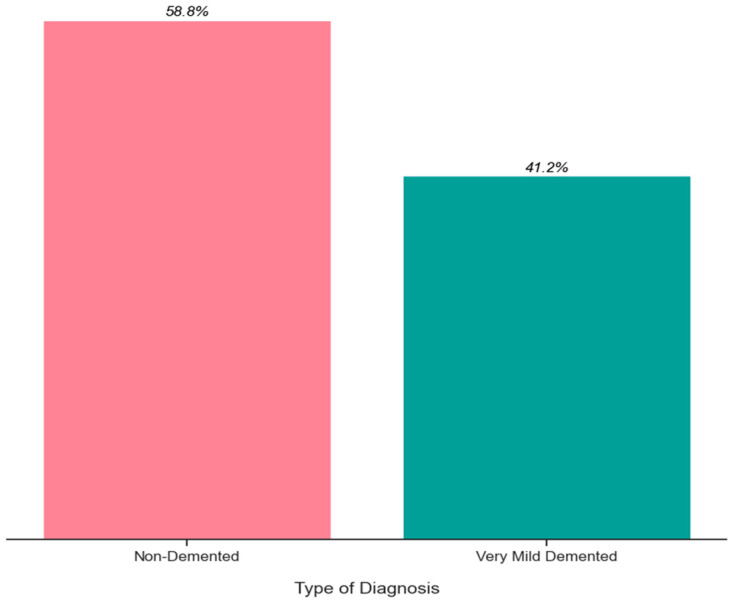
Number of samples per class.

**Figure 14 bioengineering-10-00950-f014:**
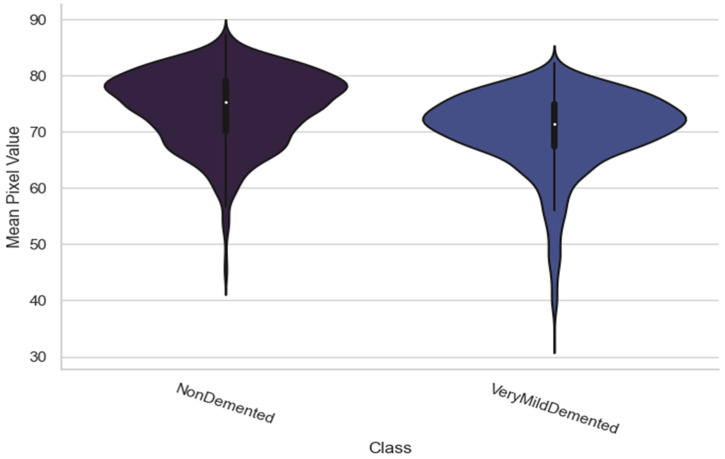
Mean value distribution for all classes.

**Figure 15 bioengineering-10-00950-f015:**
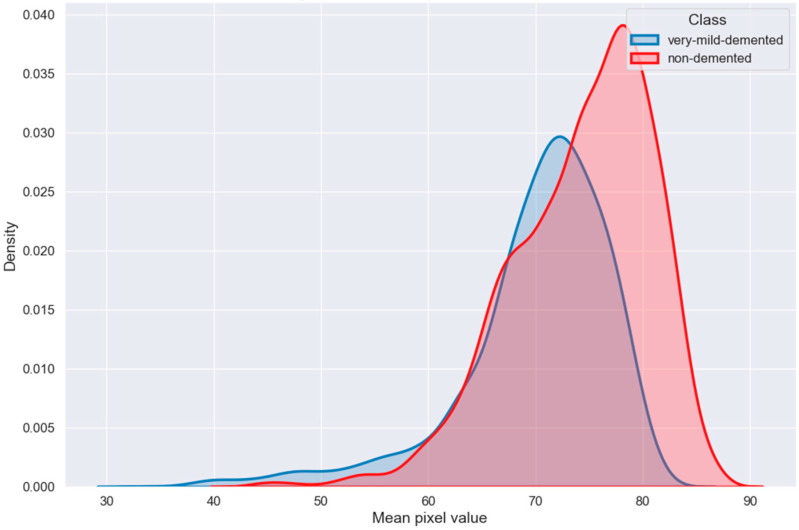
Image color mean value distribution.

**Figure 16 bioengineering-10-00950-f016:**
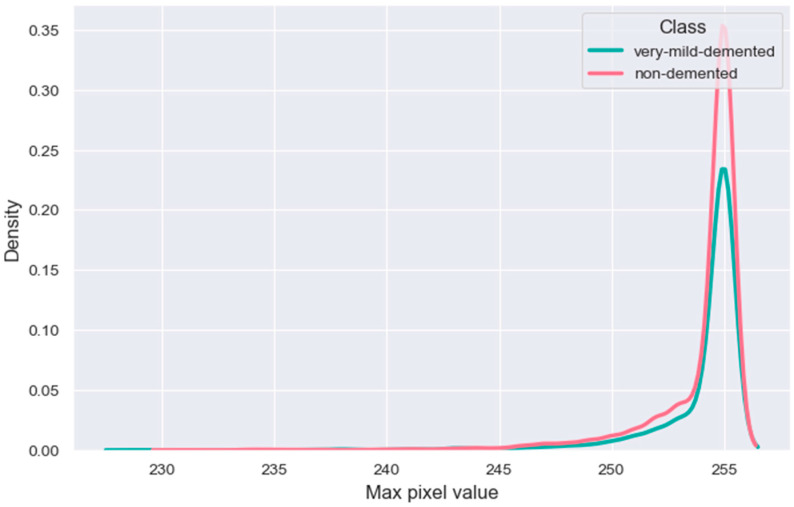
Image color max value distribution by class.

**Figure 17 bioengineering-10-00950-f017:**
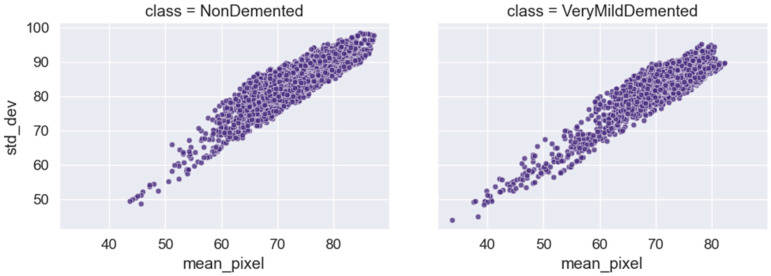
The average and variability of image samples are represented by their mean and standard deviation values.

**Table 1 bioengineering-10-00950-t001:** Recent studies for the prediction of AD.

Ref.	Approach	Methods	Dataset	Result
[10] 2018	Longitudinal analysis for AD diagnosis with RNN	MLP is first developed to learn the spatial characteristics of MR images to classify AD. The MLP outputs are then used to train an RNN with two cascaded (BGRU) layers, which, by extracting longitudinal information from the imaging data at different time intervals, generates a final categorization predicting score.	ADNI, sMRI T1-weighted 428 subjects, 198 AD patients, and 229 NC	Accuracy: 89.7%
[26] 2019	Predicting Alzheimer’s disease using LSTM	LSTM is used to predict the disease’s future state rather than to categorize it in its current state.	1105 patients are included in the MRI longitudinal time sequence data from the ADNI	AUC of AD vs. NC: 0.935, mAUC of AD vs. MCI 0.798 and mAUC of AD vs. NC vs. MCI 0.777
[30] 2019	Longitudinal analysis for diagnosis of AD with RNN	The RNN harvests longitudinal data for AD classification using three cascaded BGRU layers, whereas the CNN learns MR image spatial characteristics for classification.	ADNI 198 AD, 229 NC, and 403 MCI	Accuracy 91.33% (AD vs. NC) and 71.71% (pMCI vs. sMCI)
[31] 2020	Hybrid-model-based amalgamation for AD detection	An ensemble approach that uses a weighted average technique which can be employed to merge these models, including CNN, RNN, and LSTM.	Open Access Series of Imaging Studies (OASIS) OASIS dataset-1 for CNN and OASIS dataset-2 for RNN, LSTM	Ensemble of bagged models accuracy 92.22%, Ensemble of primary models 89.75%
[12] 2021	An end-to-end 3D-ConvLSTM for early detection of AD	With the help of high-resolution whole-brain sMRI data, this project seeks to develop or build a comprehensive 3D ConvLSTM-based framework for the early identification of AD.	OASIS-3 and ADNI 1-Screening	Accuracy: 86%, Specificity: 74%, Sensitivity: 96%, F1-score: 88% and AUC of 93%
[32] 2022	An LSTM biomarker-based prediction for AD	After 6, 12, 21, 18, 24, and 36 months, the model can predict the biomarkers (feature vectors) of a patient. These predicted biomarkers will go through layers of a neural network that are all connected. The NN layers will then decide if these biomarkers belong to a person with AD or MCIx.	ADNI, 805 subjects MRI T1-weighted	Accuracy: 88.24%

**Table 2 bioengineering-10-00950-t002:** Dataset distribution.

Class	Training	Testing
NC	2560	640
AD	1792	448
Total	4352	1088

**Table 3 bioengineering-10-00950-t003:** Parameters.

Parameter	Values
Dropout rate	0.2
Batch size	32
Cross-validation	Stratified Shuffle-Split (5 splits, 0.1 test size)
Activation function	Relu and Sigmoid
Accuracy	Metric = Accuracy
Loss function	binary_crossentropy
Optimizer	SGD (Learning Rate with Momentum)
Early stopping	Early Stopping with Epoch = 200

**Table 4 bioengineering-10-00950-t004:** Comparison of performance results for different models.

Technique	Dataset	Performance
Longitudinal analysis for AD diagnosis with RNN	ADNI	Accuracy: 89.7%
Predicting Alzheimer’s disease using LSTM	ADNI	AUC of AD vs. NC: 0.935
Longitudinal analysis for diagnosis of AD with RNN	ADNI	Accuracy: 91.33% (AD vs. NC)
Hybrid model-based amalgamation for AD detection	OASIS	Accuracy: 92.22%
A 3D-ConvLSTM end-to-end for AD early detection	OASIS +ADNI	AUC of 93%
An LSTM biomarker-based prediction for AD	ADNI	Accuracy: 88.24%
Proposed approach	Kaggle Dataset	Accuracy: 98.62%

## Data Availability

Data will be made available on request only due to ethical restrictions.

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
