# Peer review of "An Approach to Binary Classification of Alzheimer’s Disease Using LSTM"

_bioengineering, 2023, doi:10.3390/bioengineering10080950_

Round 1

Reviewer 1 Report

The approach is interesting, but some aspects need to be better addressed and further explained. Below my specific comments:

- 

-        I suggest rephrasing sentences at lines 57-58, 125, 270-271, 279, and 341-342.

-        The introduction is well organized, the topic is well contextualized, and the aim is clear.

-        Instead of “In the diagram shown below” at line 135, I suggest clearly recalling the figure.

-        Can the authors insert the citation for Kaggle Repository?

-        I encourage revising sentence at lines 233-235. First of all, the sentence should be rephrased and possibly mentioning of Table 1 should be Table 2.

-        In section 4.1 I suggest giving more details about the populations analyzed. Actually, it is not described.

-        In figure 3, I suggest specifying the modality of connection between “LSTM Model Building” and “Model Evaluation”, as well as between “Saved Model” and “Model Deployment”.

-        Separating the data into training and testing, is the reciprocal proportion (in number) of non-demented and very-mild-demented populations maintained?

-        Can the authors comment on the interpretability of the implemented model?

-        How do the explain the behavior around 175 epochs in figure 4 (and analogously in figure 5)?

-        Can the authors give a more detailed description of figures 10, 12 and 13? The last figures (10 to 15) of the manuscript are inserted but not commented.

-        At line 414, the reference to figure 24 should be reference to figure 14.

-        What different information can be deduced from figure 14 and figure 15?

-        Is the web application already available? Do the authors think that the web application could be useful for clinicians or for the general population?

There are some typos to be checked.

Reviewer 2 Report

This paper is about An Approach to Binary Classification of Alzheimer’s Disease 2 Using Long-Short-Term-Memory. However, the novelty of this approach is not clearly stated, and the paper has several weaknesses that need to be addressed. In particular:

1. The abstract should be revised to provide a more structured and informative summary of the proposed model and its results, highlighting the main contributions and novelty of the work.

2. The introduction should provide a more comprehensive review of the literature, including recent developments in deep learning methods for brain disorder diagnosis, such as ASD and SZ. Some relevant references are suggested for this purpose, but the authors should also consider other sources and provide a critical evaluation of the state of the art in this field. Also, I recommended some references that you can used some references such as https://doi.org/10.1016/j.inffus.2022.12.010 and  https://doi.org/10.1007/s11571-022-09897-w for this section.

3. The paper should include a section on "Related Works" that summarizes and compares the main studies and approaches in this area, using a tabular format that provides key information on the works, datasets, preprocessing, methods, and performance metrics. This section should also include some critical analysis and discussion of the strengths and limitations of the existing methods.

4. The proposed method should be described in more detail, including the hyperparameters and their values, to facilitate reproducibility and comparison with other methods.

5. The learning curves and loss curves of the proposed model should be presented and analyzed, to give readers a better understanding of the training process and the convergence properties of the algorithm.

6. Please plot confusion matrices. 

7. The authors should perform some baseline experiments and compare their results with those of other basic models, to demonstrate the effectiveness and generalizability of their approach.

8. The research questions should be formulated more clearly and explicitly, to guide the analysis and interpretation of the results.

9. The initial hypothesis should be stated more precisely, and its relationship to the research questions should be clarified.

10. The performance metrics should be summarized in a table that provides a comprehensive and easy-to-read overview of the main results.

11. The discussion section should critically evaluate the findings and relate them to the research questions and hypothesis, highlighting the strengths and weaknesses of the proposed method and comparing it with other approaches.

12. The main findings should be identified and discussed in terms of their novelty and contribution to the field, emphasizing the potential clinical implications and future research directions.

13. A recap of all the relevant parameters and their meaning should be added to help readers understand and replicate the experiments.

14. The clinical significance of the findings should be highlighted, explaining how they could improve the diagnosis and treatment of brain disorders related to fatigue.

15. The limitations of the study should be acknowledged and discussed. 

17. The conclusion section should provide a more detailed and insightful discussion of the implications and potential impact of the work, suggesting some concrete avenues for further research and development.

18. A comparison table should be added to the conclusion section, summarizing the main features and performance of the proposed method and other related works.

19. The English language should be improved to eliminate errors and enhance clarity and coherence, using appropriate grammar, syntax, and vocabulary.

The English language should be improved to eliminate errors and enhance clarity and coherence, using appropriate grammar, syntax, and vocabulary.

Reviewer 3 Report

The paper describes an application of a binary classifier, based on the LSTM architecture, to classify MRI of brains into "non-demented" and "very-mild-demented".

This reviewer recognizes the importance of early diagnosis of Alzheimer disease and the contributions that Deep Learning-based techniques can give in this field. However, the presented research lacks the necessary soundness and novelty to be considered for publication in a scientific journal.

Specifically

1. The related works focus on RNNs and LSTM  only, but there are other models that demononstrated their accuracy in the same application domains such as, 3D CNN and ConvLSTM (and authors included this kind of architecture in the references, such as [13], [12], and [34]). Which are the advantages of their classifier with respect to the other classifiers available in literature? The related works section should focus on such question, rather than just describing some other papers.

2. The experiments should include data used in other experiments such as those mentioned in the related works section (using, for example, brain MRIs from the ADNI and OASIS datasets), for a fair comparison. Moreover, the authors do not provide any detail about the dataset they found in Kaggle, not even the link. Data is not described. Did the authors use all the slices of each MRI? From which scanners do the RMIs come from? What about patients' age range and sex? Did the authors apply any kind of data pre-preprocessing, such as intensity normalization, cropping, scan registration, brain segmentation, etc? The data used in the experiments should be described with more details.

3. The authors should provide more details about the proposed classifier. Not all the hyperparameters are provided in the text. Some are given in the "Model Building and Deployement Flow" which, however, includes just some copy and paste of the Keras functions. For example, which was the batch size? For how many epochs did the training last? How did the authors select the hyperparameters? Did they apply a grid search strategy?

4. Did the authors evaluate the use of an early stopping strategy (for example on the validation loss) during training? In facts, from figure 4 and 5 it seems that the model is overfitting on training data. Moreover, between epoch 150 and 175 the accuracy suddenly drops (and the loss increases). Why is that, according to the authors?

5. The experiment is based on a single data split between training and test. Using a cross-validation strategy would help in getting more general results. Moreover, using the accuracy as the only performance metric is not enough. For example, AUC and ROC would allow understanding the generalization capability of the classifier. Moreover, computing Sensitivity and Specificity over the folds of a cross-validation would allow understanding the capability of the classifier in identifying both classes, more than the confusion matrix of a single split, as the authors did.

Round 2

Reviewer 1 Report

The paper has improved, but some aspects still need to be better addressed. Below my specific comments:

-        I suggest revising and rephrasing sentence at line 128 (“By fixing the RNNs' exploding and vanishing gradient issues”), 350-351 (“the input would 350 100 x 100 shape which means 100 sequences of 100 elements”; possibly a word is missing?)

-        Pay attention to the numbering of paragraphs as there are some typos.

-        In paragraph 4.1 the database is still not described (e.g., the age of the patients, etc.)

-        Changes of figure 3 are not clear (moreover, possibly, the lines indicated in the answers to reviewer are not correct). I cannot find figure 3 mentioned in the text. Can the authors clarify the modifications?

-        Can the authors better highlight the evaluation of the model interpretability and possibly reporting it in the answer to the reviewer?

-        Figure 12 is not mentioned and described in the text.

-        Can the authors comment on the different meaning of these two sentences that seems analogous to the reviewer? “Figure 14 showcases the average and variability of image samples, represented by their mean and standard deviation values” and “figure 15 presents the mean and standard deviation of image samples”.

-        In order to improve the comprehension of figures related to exploratory data analysis I suggest explaining the difference and reciprocal relationship between pixel (to which they refer talking about figures 11 and 13) and image samples (to which they refer talking about figures 14 and 15)

I suggest revising in particular the phrasing of certain sentences.

Reviewer 2 Report

The authors addressed my concerns, so this version of the paper can be accepted for publication.

Reviewer 3 Report

My only comment is for the final version of the manuscript: the authors should consider to include charts and similar figures as vector images instead of raster images. For example, they can include images as PDFs with embedded fonts.

Concerning my previous comments, the authors partially addressed them (leaving some for future work, such as the cross validation). Concerning the weird loss curve of their training and validation, I do not completely got the reply for the authors, which is very general. I understand it might depend on many factors, but I would just stop the training earlier.

Round 3

Reviewer 1 Report

The paper has further improved, and the authors answered to all the reviewer’s questions. One only aspect still needs clarification. In particular, I cannot find the description of figure 13: at line 585 it is mentioned but then the description seems to refer to figure 12. Please, revise line 585 to clarify this aspect.

Reviewer 3 Report

The authors addressed my previous comments. As such, I think the paper can be accepted in present form.